# Stimulation, Reduction and Compensation Growth, and Variable Phenological Responses to Spring and/or Summer–Autumn Warming in *Corylus* Taxa and *Cornus sanguinea* L.

Kristine Vander Mijnsbrugge [1,*] , Jessa May Malanguis [2] , Stefaan Moreels [1] , Arion Turcsán [3] and Eduardo Notivol Paino [4]

1   Department of Forest Ecology and Management, Research Institute for Nature and Forest, 9500 Geraardsbergen, Belgium; stefaan.moreels@inbo.be
2   Division of Natural Sciences and Mathematics, University of the Philippines Visayas Tacloban College, Tacloban City 6500, Philippines; malanguisjessamay@gmail.com
3   Bavarian State Institute of Forestry, Department Soil and Climate, Hans-Carl-von-Carlowitz-Platz 1, D-85354 Freising, Germany; raup25@gmail.com
4   Department of Environment, Agricultural and Forest Systems, Centro de Investigación y Tecnología Agroalimentaria de Aragón (CITA), Avda. Montañana 930, 50059 Zaragoza, Spain; enotivol@cita-aragon.es
*   Correspondence: kristine.vandermijnsbrugge@inbo.be

**Abstract:** Understanding species-specific responses to climate change allows a better assessment of the possible impact of global warming on forest growth. We studied the responses of the shrub species *Corylus avellana* L., *Corylus maxima* Mill. and intermediate forms, together stated as the *Corylus* taxa, and *Cornus sanguinea* L. upon periodically elevated temperatures in spring and/or in summer–autumn. Experiments were performed in a common garden, with Belgian and Pyrenean provenances for *Corylus avellana* and *Cornus sanguinea*. In the *Corylus* taxa, a warmer spring resulted in a reduction in height and diameter growth. Remarkably, the reduced diameter increment was restored with full compensation in the following year. The height increment for *Cornus sanguinea* was larger upon a warmer summer–autumn, concurring with a later leaf senescence. Our results suggest that *Corylus* is more sensitive to spring warming, influencing growth negatively, whereas *Cornus* is more sensitive to summer–autumn warming, influencing height growth positively. These deviating responses can be explained, at least partly, by their diverging ecological niches, with the *Corylus* taxa being more shade-tolerant compared to *Cornus sanguinea*. The warm spring conditions advanced bud burst in all studied taxa, whereas the warm summer–autumn advanced leaf senescence but prolonged its duration in the *Corylus* taxa, as well as delayed this phenophase in *Cornus sanguinea*. Little to no after-effects of the temperature treatments were detected. Although *Corylus avellana* and *Cornus sanguinea* plants originated from similar origins, their growth and phenological responses in the common garden diverged, with *Corylus* being more stable and *Cornus* displaying more variation between the Belgian and Pyrenean provenances.

**Keywords:** red dogwood; hazel; filbert; provenance trial; periodic warming; elevated temperature; bud burst; leaf senescence; common garden; cumulative logistic regression

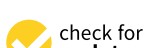



## 1. Introduction

Climate change raises global temperatures, thus influencing ecosystem processes [1]. On average, the world has already warmed 1.1 °C, affecting natural ecosystems in Europe [2]. Warming will decrease suitable habitat space for current terrestrial ecosystems and irreversibly change their composition [2]. Understanding plant responses to warming is a prerequisite for accurate predictions of the putative impact of global change on natural ecosystems, justifying the study of plant responses to elevated temperatures. Warming can

stimulate net photosynthesis in woody plants [3,4] and can shift their phenophases [5–7]. Alterations in the length of the growing season due to climate change, together with predicted range shifts, can substantially affect the dynamics in ecosystems, including carbon and water cycling [8–10].

Phenological traits are amongst the most responsive characteristics of plants to global warming [11]. The timing of bud burst and leaf senescence mark the beginning and the end of the growing season in temperate deciduous trees species and balances between the risks of damage due to late frosts in spring or early frosts in autumn and the benefits of a prolonged period of photosynthesis with more growth and production of biomass, rendering a competitive advantage [11]. The timings of bud burst and leaf senescence vary both within and among species, and, in many temperate tree and shrub species, these phenophases are highly sensitive to temperature [10,12]. Advancement or retardation of bud burst and/or leaf senescence directly adjusts the length of the growing season and can consequently alter ecosystem functioning and ecosystem structure [13,14]. In addition, phenological shifts due to climate change may not only alter vegetative phenophases but also reproductive phenophases, impacting crop production for woody species producing fruits with economic value, e.g., *Corylus* taxa [15].

Raising temperatures can allocate carbon to internal growth processes in trees, although there are limits above the temperature optima [16]. Predicting the size of growth changes upon warming is difficult because, amongst others, tree species respond differently to warming and heat stress [16] and display a broad range of optimum temperatures for photosynthesis, which may vary between provenances from different thermal origins [17,18]. Additionally, trees from temperate and boreal forests can to some extent acclimatize their temperature optima for photosynthesis, as observed in experimental warming studies [12,13]. Growth stimulation, growth retardation, or a status quo has been described for temperate tree species in warming experiments [16]. Tree populations growing near to the warm range limit displayed reduced net photosynthesis and growth upon warming, whereas populations near to the cold range limit responded positively to warming [19].

Elevated temperatures in different seasons may display variable effects on the timing and the duration of bud burst and leaf senescence, as well as on the growth, as it has been suggested that ecophysiological processes that vary along the growing season may respond in a variable way to nonuniform seasonal warming [20,21]. Limited experiments focus on the responses of tree species to seasonally variable warming [21].

*Corylus avellana* L. and *Cornus sanguinea* L. are shrub species occurring naturally in forests and woody landscape elements in the cultural landscape in large parts of Europe, preferring temperate climates [22,23]. In natural stands, they are widely distributed in Europe, ranging from Scandinavia to the south of the continent [23]. In Belgium, the species are also very common [24]. Hazel is a wind-pollinated shrub. As the kernel of the nut is edible, *Corylus avellana* has for long been selected and cultivated for larger nuts, and intermediate forms that probably originated from spontaneous crosses between cultivated and natural *Corylus avellana* have been described [25,26]. The production of edible hazelnuts is an important economic activity, e.g., in Turkey [27]. *Corylus maxima* Mill. is a nonindigenous species in Belgium, native to southeastern Europe and southwestern Asia [28], and it has been planted as an ornamental in many gardens in Belgium. *Corylus avellana* and *Corylus maxima* are interfertile, and molecular genetic analyses have indicated a history of past hybridization [29]. Although leaf morphology is very similar between the two species, nut morphology differs, and intermediate forms can be found in Belgium. Nuts are more elongated in *Corylus maxima* and fully enclosed in a tubular involucre (husk) that is at least twice the length of the nut, whereas, in *Corylus avellana*, each nut is held in a short leafy involucre which encloses about three-quarters of the nut up to the length of the nut [28]. *Cornus sanguinea* is a deciduous insect-pollinated shrub species common to most of Europe and West Asia [22,30]. The nontoxic but unpleasant tasting fruits of *C. sanguinea* are berry-like drupes, 5–8 mm wide, without any economic value [22].

To be able to assess the effect of global warming on woody vegetations, it is useful to get a better understanding of species-specific responses to warming conditions not only for the dominant and economically important tree species but also for regular shrub species. We studied the response of juvenile *Corylus avellana*, *Corylus maxima* and intermediate forms, and *Cornus sanguinea* in a temperature manipulative experiment with spring and/or summer–autumn warming. For *Corylus avellana* and *Cornus sanguinea*, a local Belgian and a nonlocal Spanish provenance were included in a common garden setting. For *Corylus maxima* and the intermediate forms, only Belgian plants were included. The experiment was conducted in greenhouse conditions. We aimed to look at relative differences in plant responses. We hypothesized that (i) spring warming, summer–autumn warming, or a combination of both would affect height and diameter growth, (ii) the timing and duration of the phenophases bud burst and leaf senescence are affected by the applied spring and/or summer–autumn warming, and (iii) differences in growth and phenology are expressed between the different *Corylus* taxa and between the local and nonlocal provenances.

## 2. Materials and Methods

### 2.1. Seed Collection and Germination

Nuts and berries from *Corylus avellana* and *Cornus sanguinea* were collected in 2016 in several natural populations, further called provenances, in Belgium (northern region, Atlantic climate) and in Spain (Pyrenees, Alpine climate) (Table 1). Nuts and berries of each mother shrub were kept apart. For the collections, undisturbed autochthonous populations were chosen. In selecting *Corylus avellana* populations, nuts were carefully chosen to ensure that small nuts with not too long involucres were collected. Nuts were also collected from *Corylus maxima*, a nonindigenous shrub in Belgium, often planted as an ornamental in gardens. Planted individuals were chosen as seed source (Table 1). Intermediate forms were determined in the field on the basis of larger sized nuts and/or larger involucres that were longer than the nut length, but that did not yet fully enclose the nut. Intermediate forms were found in Belgium growing in more anthropogenic growth sites.

**Table 1.** Description of the sampling sites and details on the seedlings of the *Corylus* taxa and *Cornus sanguinea*. Abb.: abbreviation of the provenance, including an abbreviation of the taxon for *Corylus*; $n^\circ_{mpl}$: number of mother plants; $n^\circ$: number of seedlings; $n^\circ_t$ cc/cw/wc/ww: number of seedlings conforming with the spring treatment (first letter) and the summer–autumn treatment (last letter) of 2018, with cc: cold–cold, cw: cold–warm, wc: warm–cold, and ww: warm–warm.

| Species | Country, Region, Village | Abb. | Latitude | Longitude | Altitude (m a.s.l.) | $n^\circ_{mpl}$ | $n^\circ$ | $n^\circ_t$ cc/cw/wc/ww |
|---|---|---|---|---|---|---|---|---|
| *Corylus avellana* | Belgium, Flanders | Be_Cave | 50.947929 | 3.765214 | 24 | 10 | 92 | 26/19/18/29 |
| | Spain, Pyrenees, Linas De Broto | Sp_Cave | 42.630049 | −0.169068 | 1270 | 9 | 80 | 26/14/18/22 |
| Intermediate forms | Belgium, Flanders | Be_Cx | 50.965736 | 3.693461 | 10 | 7 | 61 | 17/13/13/18 |
| *Corylus maxima* | Belgium, Flanders | Be_Cmax | 50.992723 | 3.775170 | 19 | 2 | 20 | 6/4/4/6 |
| *Cornus sanguinea* | Belgium, Flanders, Kriephoek | Be | 50.953324 | 3.663467 | 10 | 21 | 205 | 61/42/44/58 |
| | Spain, Pyrenees, Linas De Broto | Sp | 42.630049 | −0.169068 | 1270 | 12 | 117 | 34/24/24/35 |

The climate and day length at the origin of the provenances for *Corylus avellana* and *Cornus sanguinea* are shown in Figure 1 (mean monthly maximum and minimum temperature, mean monthly precipitation, for 1970–2000, WorldClim version 2 [23] and NOAA Solar Calculator [24]). *Corylus maxima* and the intermediate forms were also collected in Belgium, implying that the Belgian climate data in Figure 1 count for these two taxa as well.

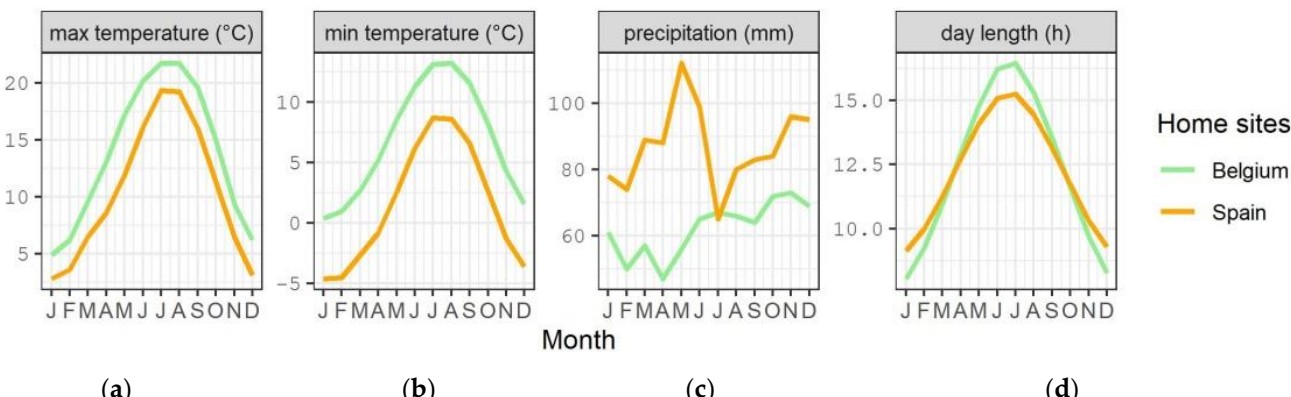

**Figure 1.** Climate and day length at the origin of the provenances of *Corylus avellana*, *Corylus maxima* and the intermediate forms, and *Cornus sanguinea*. (**a**) Mean monthly maximum temperature; (**b**) mean monthly minimum temperature; (**c**) mean monthly precipitation; (**d**) day length.

After the collection in 2016, berries from *Cornus sanguinea* were cleaned. Berries were soaked overnight in water, and seeds were manually pushed out from the berries. Then, cleaned seeds and nuts from the *Corylus* taxa were put in regular potting soil (organic matter 20%, pH 5.0–6.5, E.C. 450 μS/cm, 25% dry matter, 1.5 kg/m$^3$ NPK 12 + 14 + 24) at the Research Institute for Nature and Forest in Geraardsbergen, Belgium. In the next spring of 2017, up to 10 germinated seedlings for each mother shrub were taken for the common garden experiment (Table 1). The emerged seedlings were transferred to forestry propagation trays using the regular potting soil. The seedlings were kept in a greenhouse without side walls and were watered to full capacity twice a week. An automatic gray shade net operating in the greenhouse protected the seedlings from strong insolation.

### 2.2. Temperature Treatments

The temperature treatments performed in 2018 were similar to a warming experiment with the shrub *Prunus spinosa* L. [31]. The *Corylus* taxa and *Cornus sanguinea* were not intermingled during the experiment. At the end of 2017, the seedlings were divided into two groups (two for the *Corylus* taxa and two for *Cornus sanguinea*). For each mother plant, the derived seedlings were randomly distributed to the two groups. In each group, the seedlings were randomly intermingled (completely randomized design) and placed back in the same type of propagation trays. On 20 February 2018, the two groups of trays were transferred to two separate chambers in the greenhouse. The two chambers had a different temperature inside (Figure 2a), as one was heated and the other was not (further called warm and cold conditions). On average, there was a temperature difference of 5.6 °C between the two chambers. To keep the plants well hydrated in both chambers, they were watered to full capacity twice a week. The plants remained in the chambers until the leaves of all seedlings were unfolded. This was the case on 10 April 2018. After this treatment, the trays from the two chambers were placed together in a standard greenhouse, randomly intermingled. The seedlings of *Cornus sanguinea* were cut back at 15 cm height. The plants remained in this greenhouse until the end of July 2018.

On 1 August 2018, the plants were for the second time distributed into two new groups. From the seedlings that experienced the warm condition in the spring, half of the plants were allocated to the first group and the other half were allocated to the second group. The same was applied for the cold condition, resulting in a factorial design. The first new group remained in the standard greenhouse, while the second group was relocated to the greenhouse without walls. From August to October, the two greenhouses differed on average by 1.9 °C (Figure 2b). In both places, an automatic shade net functioned in the same way, protecting plants from harmful insolation. At the beginning of November, all plants were brought together in the greenhouse without walls.

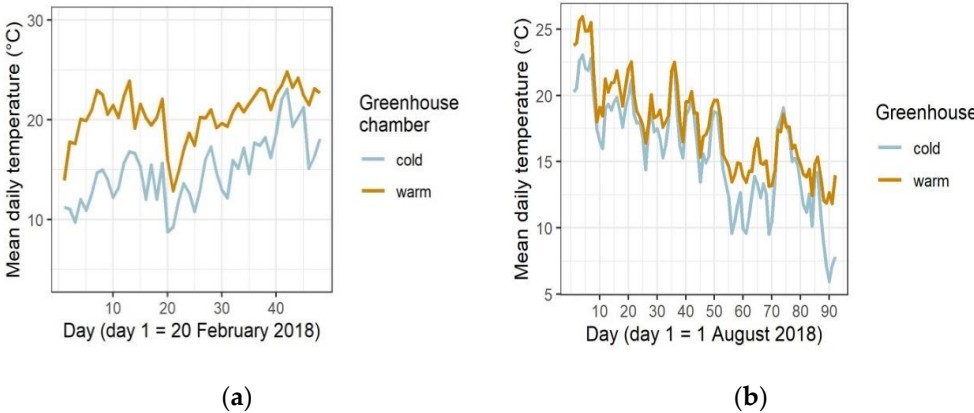

(**a**)　　　　　　　　　　　　　　　　　　　　(**b**)

**Figure 2.** Mean daily greenhouse temperatures in spring (**a**) and in summer–autumn (**b**) in 2018.

In the winter of 2018, the plants were transferred to 1 L pots using standard potting soil (organic matter 20%, pH 5.0–6.5, E.C. 450 µS/cm, 25% dry matter, 1.5 kg/m$^3$ NPK 12 + 14 + 24). The pots were intermingled and placed on a container field at the Research Institute for Agriculture, Fisheries and Food, in Melle, Belgium. An automatic watering system with rotating sprinklers supplied water to the plants every 2 days. The plants remained here during the growing season of 2019. In the winter of 2019, all plants were repotted to 4 L containers using standard potting soil. All plants were once more intermingled and stayed on the container field during the growing season of 2020. As *Cornus sanguinea* grew quickly, plants were cut back in July 2020 at 20 cm height.

### 2.3. Measurements and Phenological Observations

All described measurements and observations were performed on the *Corylus* taxa and on *Cornus sanguinea*.

In the winters of 2017, 2018, 2019, and 2020, measurements of plant height and stem diameter at 1 cm above the pot soil level were performed on all plants.

The rate of bud burst was observed regularly in the spring of 2018, 2019, and 2020 (Table 2), following a scoring system of five developmental stages: (1) buds in winter rest, (2) buds swelling, (3) green leaves protruding, but still folded, (4) leaves unfolding, and (5) leaves unfolded. The rate of leaf senescence was assessed in the autumn of 2018, 2019, and 2020 (Table 2) following a scoring system of five senescence stages: (1) green leaves, (2) light-green leaves, (3) leaves becoming yellow, (4) leaves becoming brown, and (5) brown leaves starting to fall off [32].

**Table 2.** Observation dates for bud burst and leaf senescence.

| Genus | Year | Bud Burst | Leaf Senescence |
|---|---|---|---|
| *Corylus* | 2018 | 26/2, 1/3, 5/3, 8/3, 12/3, 15/3, 19/3, 26/3, 3/4, 9/4 | 28/9, 17/10 |
| | 2019 | 6/3, 14/3, 4/4 | 16/9, 15/10, 19/11 |
| | 2020 | 9/3, 27/3, 8/4 | 14/9, 2/10, 23/10 |
| *Cornus* | 2018 | 26/2, 1/3, 5/3, 8/3, 12/3, 15/3, 19/3, 26/3, 3/4, 9/4 | 28/9, 16/10 |
| | 2019 | 5/4, 10/4, 17/4 | 12/9, 15/10, 19/11 |
| | 2020 | 9/3, 16/3, 27/3, 6/4 | 21/9, 12/10, 6/11 |

### 2.4. Statistical Analysis

Applying the open-source software R version 3.6.1, (generalized) linear mixed models were fitted [33]. The structure of the models was similar for the *Corylus* taxa and for *Cornus sanguinea*. We analyzed the growth and the phenology of the plants in response to the temperature treatments in 2018. The temperature treatment (T) in the greenhouse chambers in the spring of 2018 consisted of two conditions: "cold" and "warm" (Tc and Tw). This two-condition variable was present in the fixed part of the model for bud burst in 2018. For all

other response variables, including bud burst in 2019 and 2020, the temperature treatments in 2018 (fixed part) comprised four conditions: cold in spring and cold in summer–autumn ("cold–cold", Tcc), cold in spring and warm in summer–autumn ("cold–warm", Tcw), warm in spring and cold in summer–autumn ("warm–cold", Twc) and warm in both spring and summer–autumn ("warm–warm", Tww). "Cold–cold" was the standard treatment to which the others were compared. Provenance (P) was present in the fixed part of the models. For *Cornus sanguinea*, the provenance "Belgium" was the standard provenance to which the Spanish provenance was compared. For the *Corylus* taxa, provenance (P) included not only a geographic aspect (Belgian and Spanish provenances for *Corylus avellana*) but also a taxonomic aspect (different taxa for Belgium: the intermediate forms and *Corylus maxima*). In this case, the Belgian provenance of *Corylus avellana* was also the standard provenance.

Models for plant height and stem diameter were fitted applying linear mixed models in the package lme4 [34].

Plants were scored for the phenological development at different dates. Therefore, the day of scoring (D) was present in the fixed part of the phenological models. The height of the plants was also added in the fixed part of the models with $H_1$ (height at the end of 2017) up to $H_4$ (height at the end of 2020). An interaction term between treatment and day was added in the fixed part of the models for bud burst and leaf senescence in 2018. This interaction term accounted for the different duration in time needed for the plants to accomplish the total phenophase during the warming experiments in spring and summer–autumn of 2018.

The random part of the models included a unique identity code for each mother plant. When needed, an individual unique plant identity code was additionally included to account for several observations at different dates on the same plants.

Plant height and stem diameter ($H_1$ and $D_1$) at the end of the first growing season, before the temperature treatments (2017), were calculated as follows:

$$H_1 = \alpha_{H_1} + \beta_{PH_1} P, \tag{1}$$

$$D_1 = \alpha_{D_1} + \beta_{PD_1} P. \tag{2}$$

The growth increments of both plant height and stem diameter at the end of 2018 ($Hi_2$ and $Di_2$) were calculated as follows:

$$Hi_2 = \alpha_{Hi_2} + \beta_{PHi_2} P + \beta_{THi_2} T + \beta_{H1Hi_2} H_1, \tag{3}$$

$$Di_2 = \alpha_{Di_2} + \beta_{PDi_2} P + \beta_{TDi_2} T + \beta_{H1Di_2} D_1. \tag{4}$$

The growth increments of plant height and stem diameter at the end of 2019 and 2020 followed the same structure as the increment models of 2018, with starting height $H_2$ or $D_2$ and $H_3$ or $D_3$, respectively, instead of $H_1$ or $D_1$.

The phenological scorings belonged to an ordinal data type comprising ordered levels with unknown distances between the levels. This type of data can be modeled with cumulative logistic regression in the R package ordinal [35]. The function "clmm" fits cumulative link mixed models. The cumulative probability (p) is the chance to have reached a given level of the phenological variable or a level below, i.e., the chance to have maximally reached the given level. The bud burst scorings were ordered from the end of the phenophase to the start: from unfolded leaves to buds in winter rest (from 5 to 1). The scorings for leaf senescence were ordered in a normal chronological way, from green leaves to brown leaves that are falling off (from 1 to 5). The order of the bud burst scorings was reversed for an easier comprehension of the modeled probabilities. A probability of having reached maximally a scoring of, e.g., 3 in a bud burst variable that is ordered from 5 to 1 is the chance of having reached a scoring of 5, 4, or 3. Thus, plants displaying an early bud burst (having a higher score at a given time) have a higher modeled probability of having reached a score of 5, 4, or 3.

Bud burst in 2018 was calculated as follows:

$$\log(p_1/(1 - p_1)) = \alpha_{Trp1} - \beta_{Pp1}P - \beta_{Dp1}D - \beta_{Tp1}T - \beta_{DTp1}DT - \beta_{H1p1}H_1, \quad (5)$$

where $\alpha_{Trp1}$ is an estimated threshold value for the passing on from one level of the bud burst variable to the next. A significant interaction term between the day of observation and the temperature treatment indicated that the duration of the bud burst process differed between the warm and the cold condition.

Leaf senescence in 2018 had the same model structure as bud burst in 2018, but with the replacement of plant height at the end of 2017 ($H_1$) by plant height at the end of 2018 ($H_2$). Furthermore, the treatment variable (T) now consisted of four conditions instead of two ("cold–cold", "cold–warm", "warm–cold", and "warm–warm").

Bud burst in 2019 was calculated as follows:

$$\log(p_2/(1 - p_2)) = \alpha_{Trp2} - \beta_{Pp2}P - \beta_{Dp2}D - \beta_{Tp2}T - \beta_{H2p2}H_2. \quad (6)$$

Leaf senescence in 2019 and bud burst in 2020 had the same model structure as bud burst in 2019, but with the replacement of plant height at the end of 2018 ($H_2$) by plant height at the end of 2019 ($H_3$). Leaf senescence in 2020 also had the same model structure, but with plant height at the end of 2020 ($H_4$).

## 3. Results

### 3.1. Temperature Treatments Affect the Growth

For the *Corylus* taxa, the seedlings that experienced the cold spring and warm summer–autumn condition in 2018 ("cold–warm", Tcw in Table 3) displayed a larger height increment ($p = 0.004$, Table 3) and a higher diameter increment ($p < 0.001$, Table 3) in this year compared to the cold spring and cold summer–autumn condition ("cold–cold", Tcc). The seedlings that experienced the "warm–cold" (Twc) and the "warm–warm" (Tww) conditions displayed a twofold lower height ($p < 0.001$, Table 3) and a twofold lower diameter increment in 2018 ($p < 0.001$, Table 3) in comparison with the "cold–cold" condition (Figure 3a,b). The height increment in 2019 and in 2020 did not differ among the seedlings that experienced the different temperature treatments (no significant *p*-values, Table 3). On the contrary, the diameter increment in 2019 was higher for the seedlings in the "warm–cold" and "warm–warm" conditions ($p = 0.001$ and $p < 0.001$, respectively, Table 3). Thus, the conditions "warm–cold" and "warm–warm" in 2018 resulted in a lower diameter increment, but this was compensated for by a larger diameter increment for these two conditions in 2019 (Figure 3b). In 2020, there was no longer a significant difference in diameter increment between the different temperature conditions (no significant *p*-values, Table 3).

The height increment in 2018 for the seedlings of *Cornus sanguinea* was significantly larger in the "warm–warm" condition and the "cold–warm" condition in comparison to the "cold–cold" condition (both $p < 0.001$, Table 3, Figure 3c). A significantly larger diameter increment in 2018 was only detected for the "cold–warm" condition in comparison to the "cold–cold" condition ($p < 0.001$, Table 3, Figure 3d). Although in 2019 the influence of the treatments during 2018 disappeared for the height increment (no significant *p*-values, Table 3), the diameter increment in this year was still larger for the "cold–warm" treatment ($p < 0.001$, Table 3, Figure 3d), and a small effect was still present even the year after, in 2020 ($p = 0.048$, Table 3).

**Table 3.** Estimates and *p*-values for the modeled height and diameter increments for the years 2018 until 2020 for the *Corylus* taxa and for *Cornus sanguinea*. For the *Corylus* taxa, the standard taxon/region is the Belgian *Corylus avellana* to which the Belgian intermediate forms (Be_Cx), the Belgian *Corylus maxima* (Be_Cmax), and the Spanish *Corylus avellana* (Sp_Cave) are compared. For *Cornus sanguinea*, the Belgian provenance is the standard provenance to which the Spanish provenance (Sp) is compared. The "cold–cold" condition in the temperature treatments during 2018 is the standard to which the other three conditions "cold–warm", "warm–cold", and "warm–warm" are compared (Tcw, Twc, and Tww, respectively). H17/D17 to H19/D19 indicates that, in the models for the height increments, the starting height is present (H17 until H19), whereas, for the models of the diameter increments, the starting diameter is present (D17 until D19).

| | | *Corylus* Taxa | | | | *Cornus sanguinea* | | | | |
| | | Height Increment | | Diameter Increment | | | Height Increment | | Diameter Increment | |
| Year | Variable | Estimate | *p*-Value | Estimate | *p*-Value | Variable | Estimate | *p*-Value | Estimate | *p*-Value |
|---|---|---|---|---|---|---|---|---|---|---|
| 2018 | (Intercept) | 12.41 | <0.001 *** | 2.20 | <0.001 *** | (Intercept) | 17.68 | <0.001 *** | 2.76 | <0.001 *** |
| | Tcw | 3.84 | 0.004 ** | 0.23 | <0.001 *** | Tcw | 15.10 | <0.001 *** | 0.27 | <0.001 *** |
| | Twc | −8.78 | <0.001 *** | −0.32 | <0.001 *** | Twc | −1.04 | 0.487 | 0.03 | 0.698 |
| | Tww | −4.85 | <0.001 *** | −0.30 | <0.001 *** | Tww | 7.71 | <0.001 *** | 0.00 | 0.951 |
| | H17/D17 | 0.02 | 0.630 | −0.47 | <0.001 *** | H17/D17 | 0.28 | <0.001 *** | −0.54 | <0.001 *** |
| | Be_Cx | 2.55 | 0.096 | 0.34 | 0.025 * | Sp | 1.95 | 0.265 | −0.10 | 0.111 |
| | Be_Cmax | 2.85 | 0.204 | 0.53 | 0.026 * | | | | | |
| | Sp_Cave | −1.03 | 0.454 | −0.15 | 0.279 | | | | | |
| 2019 | (Intercept) | 5.02 | <0.001 *** | 2.40 | <0.001 *** | (Intercept) | 4.18 | <0.001 *** | 2.78 | <0.001 *** |
| | Tcw | −0.43 | 0.469 | −0.10 | 0.312 | Tcw | 0.01 | 0.979 | 0.43 | <0.001 *** |
| | Twc | −0.31 | 0.604 | 0.35 | 0.001 ** | Twc | −0.57 | 0.178 | 0.12 | 0.245 |
| | Tww | 0.23 | 0.661 | 0.34 | <0.001 *** | Tww | 0.31 | 0.437 | 0.17 | 0.073 |
| | H18/D18 | −0.05 | 0.007 ** | −0.34 | <0.001 *** | H18/D18 | −0.02 | 0.190 | −0.39 | <0.001 *** |
| | Be_Cx | 0.60 | 0.344 | 0.02 | 0.885 | Sp | −1.05 | 0.047 * | −0.13 | 0.143 |
| | Be_Cmax | 1.43 | 0.131 | 0.32 | 0.124 | | | | | |
| | Sp_Cave | −0.15 | 0.797 | −0.01 | 0.951 | | | | | |
| 2020 | (Intercept) | 7.25 | <0.001 *** | 4.31 | <0.001 *** | (Intercept) | 59.3 | <0.001 *** | 4.78 | <0.001 *** |
| | Tcw | −0.66 | 0.257 | −0.16 | 0.294 | Tcw | −1.07 | 0.721 | 0.36 | 0.048 * |
| | Twc | −0.12 | 0.832 | −0.22 | 0.135 | Twc | 1.05 | 0.694 | 0.05 | 0.771 |
| | Tww | 0.23 | 0.664 | −0.07 | 0.604 | Tww | −3.11 | 0.215 | 0.09 | 0.552 |
| | H19/D19 | −0.08 | <0.001 *** | −0.17 | 0.035 * | H19/D19 | 0.09 | 0.340 | −0.34 | <0.001 *** |
| | Be_Cx | 0.16 | 0.803 | 0.08 | 0.696 | Sp | −10.47 | <0.001 *** | −0.24 | 0.184 |
| | Be_Cmax | −0.36 | 0.716 | −0.08 | 0.796 | | | | | |
| | Sp_Cave | −0.81 | 0.184 | −0.25 | 0.169 | | | | | |

*** $p < 0.001$; ** $p < 0.01$; * $p < 0.05$.

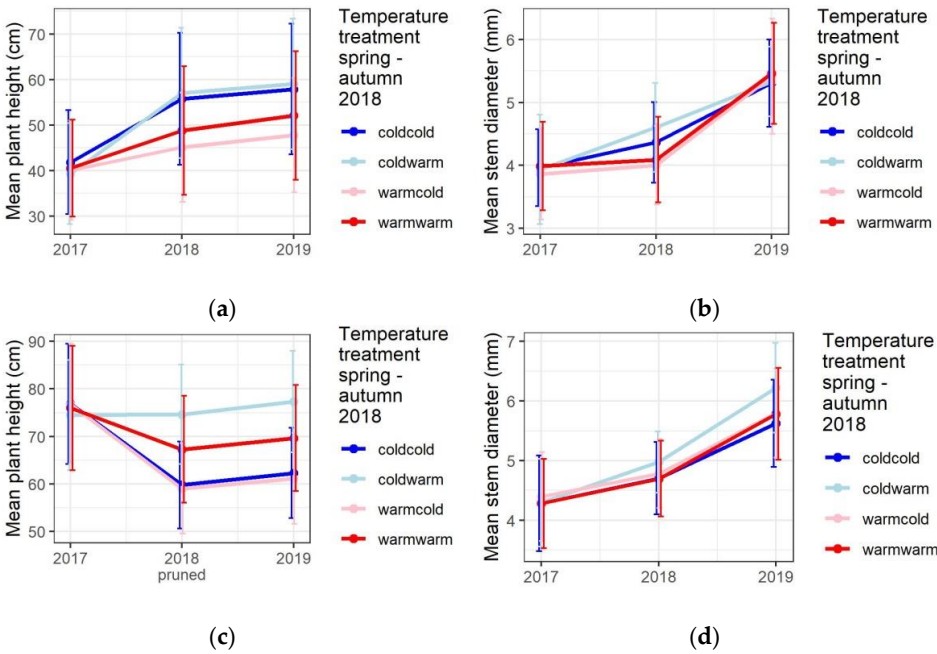

(a)　　　　　　　　　　　　　　　　　(b)

(c)　　　　　　　　　　　　　　　　　(d)

**Figure 3.** Mean and standard deviation of height (**a**,**c**) and diameter (**b**,**d**) measurements of the *Corylus* taxa (**a**,**b**) and *Cornus sanguinea* (**c**,**d**) according to the temperature treatments in 2018.

### 3.2. Temperature Treatments Affect Phenology

In the first temperature treatment in the spring of 2018, with two conditions "cold" and "warm", bud burst started earlier ($p < 0.001$ for the "warm" condition in the *Corylus* taxa and *Cornus sanguinea*, Table 4). Moreover, the duration of this phenophase was shorter in the "warm" condition compared to the "cold" condition ($p < 0.001$ for the interaction term between day and "warm" condition in the *Corylus* taxa and *Cornus sanguinea*, Table 4, Figure 4a,d).

For the seedlings from the *Corylus* taxa that experienced the warmer spring condition, leaf senescence in 2018 started earlier ($p = 0.04$ for "warm–cold" and $p < 0.001$ for "warm–warm" condition, Table 4). In addition, the duration of the senescence phenophase was longer in the seedlings that experienced the warm summer–autumn condition ($p < 0.001$ for the interaction term between day and "cold–warm", $p = 0.018$ for the interaction term between day and "warm–warm", Table 4). As shown in Figure 4b, the "cold–warm" and the "warm–warm" conditions displayed a less steep slope. Few after-effects of the temperature treatments in 2018 were observed in the *Corylus* taxa in 2019. Only the seedlings that experienced the "cold–warm" condition burst their buds later in the spring of 2019 ($p < 0.001$ for "cold–warm", Table 4, Figure 4c). An after-effect was no longer detected in the timing of the leaf senescence in 2019 or in the timing of bud burst or leaf senescence in 2020 (no significant $p$-values for the temperature treatments, Table 4).

**Table 4.** Estimates and $p$-values for the modeled bud burst and leaf senescence for the years 2018 until 2020 for the *Corylus* taxa and for *Cornus sanguinea*. For the *Corylus* taxa, the standard taxon/region is the Belgian *Corylus avellana* to which the Belgian intermediate forms (Be_Cx), the Belgian *Corylus maxima* (Be_Cmax), and the Spanish *Corylus avellana* (Sp_Cave) are compared. For *Cornus sanguinea*, the Belgian provenance is the standard provenance to which the Spanish provenance (Sp) is compared. The "cold–cold" condition in the temperature treatments during 2018 is the standard to which the other three conditions "cold–warm", "warm–cold", and "warm–warm" are compared (Tcw, Twc, and Tww, respectively). For the bud burst in 2018, there were only two conditions, with the "cold" condition as the standard to which the "warm" condition is compared (Tw). H17 to H19 are the plant heights in 2017 to 2019. D is the day of observation.

| | | *Corylus* Taxa | | | | | | | *Cornus sanguinea* | | | | | |
| | | Bud Burst | | | Leaf Senescence | | | Bud Burst | | | Leaf Senescence | | |
| Year | Variable | Estimate | *p*-Value | Variable | Estimate | *p*-Value | Variable | Estimate | *p*-Value | Variable | Estimate | *p*-Value |
|---|---|---|---|---|---|---|---|---|---|---|---|---|
| 2018 | D | −0.50 | <0.001 *** | D | 0.30 | <0.001 *** | D | −0.26 | <0.001 *** | D | 0.47 | <0.001 *** |
| | Tw | 1.39 | <0.001 *** | Tcw | 0.25 | 0.540 | Tw | 1.55 | <0.001 *** | Tcw | −2.30 | 0.002 ** |
| | | | | Twc | 0.88 | 0.040 * | | | | Twc | 0.24 | 0.692 |
| | | | | Tww | 1.47 | <0.001 *** | | | | Tww | −2.22 | 0.001 ** |
| | D:Tw | −0.69 | <0.001 *** | D:Tcw | −0.11 | <0.001 *** | D:Tw | −0.27 | <0.001 *** | D:Tcw | −0.03 | 0.496 |
| | | | | D:Twc | 0.05 | 0.069 | | | | D:Twc | −0.01 | 0.768 |
| | | | | D:Tww | −0.06 | 0.018 * | | | | D:Tww | −0.03 | 0.438 |
| | H17 | 0.15 | <0.001 *** | H18 | 0.04 | <0.001 *** | H17 | −0.01 | 0.583 | H18 | 0.05 | 0.023 * |
| | Be_Cx | −1.45 | 0.135 | Be_Cx | −0.12 | 0.698 | Sp | −0.59 | 0.017 * | Sp | −0.68 | 0.168 |
| | Be_Cmax | −3.95 | 0.008 ** | Be_Cmax | 0.72 | 0.103 | | | | | | |
| | Sp_Cave | 1.18 | 0.190 | Sp_Cave | 0.63 | 0.026 * | | | | | | |
| 2019 | D | −0.22 | <0.001 *** | D | 0.14 | <0.001 *** | D | −0.81 | <0.001 *** | D | 0.16 | <0.001 *** |
| | Tcw | 1.55 | <0.001 *** | Tcw | 0.35 | 0.191 | Tcw | 1.20 | 0.001 ** | Tcw | 0.26 | 0.316 |
| | Twc | −0.86 | 0.004 ** | Twc | −0.18 | 0.519 | Twc | −0.27 | 0.405 | Twc | 0.41 | 0.077 |
| | Tww | 0.62 | 0.015 * | Tww | 0.30 | 0.227 | Tww | 1.05 | <0.001 *** | Tww | 0.01 | 0.949 |
| | H18 | 0.06 | <0.001 *** | H19 | 0.00 | 0.886 | H18 | −0.02 | 0.111 | H19 | 0.00 | 0.811 |
| | Be_Cx | −1.05 | 0.034 * | Be_Cx | 0.26 | 0.350 | Sp | −1.42 | <0.001 *** | Sp | 0.49 | 0.008 ** |
| | Be_Cmax | −2.83 | <0.001 *** | Be_Cmax | −0.02 | 0.953 | | | | | | |
| | Sp_Cave | 0.29 | 0.522 | Sp_Cave | 0.12 | 0.658 | | | | | | |
| 2020 | D | −0.40 | <0.001 *** | D | 0.39 | <0.001 *** | D | −0.75 | <0.001 *** | D | 0.13 | <0.001 *** |
| | Tcw | −0.57 | 0.174 | Tcw | −0.58 | 0.110 | Tcw | 0.24 | 0.622 | Tcw | −0.11 | 0.744 |
| | Twc | −0.03 | 0.941 | Twc | 0.34 | 0.344 | Twc | −0.37 | 0.388 | Twc | 0.29 | 0.373 |
| | Tww | 0.01 | 0.981 | Tww | −0.07 | 0.821 | Tww | −0.09 | 0.813 | Tww | 0.04 | 0.885 |
| | H19 | 0.04 | 0.001 ** | H20 | 0.05 | <0.001 *** | H19 | 0.007 | 0.652 | H20 | −0.006 | 0.382 |
| | Be_Cx | −1.18 | 0.020 * | Be_Cx | 0.85 | 0.010* | Sp | −2.08 | <0.001 *** | Sp | −0.47 | 0.093 |
| | Be_Cmax | −2.24 | 0.003 ** | Be_Cmax | 0.34 | 0.482 | | | | | | |
| | Sp_Cave | 0.18 | 0.688 | Sp_Cave | −0.01 | 0.983 | | | | | | |

*** $p < 0.001$; ** $p < 0.01$; * $p < 0.05$.

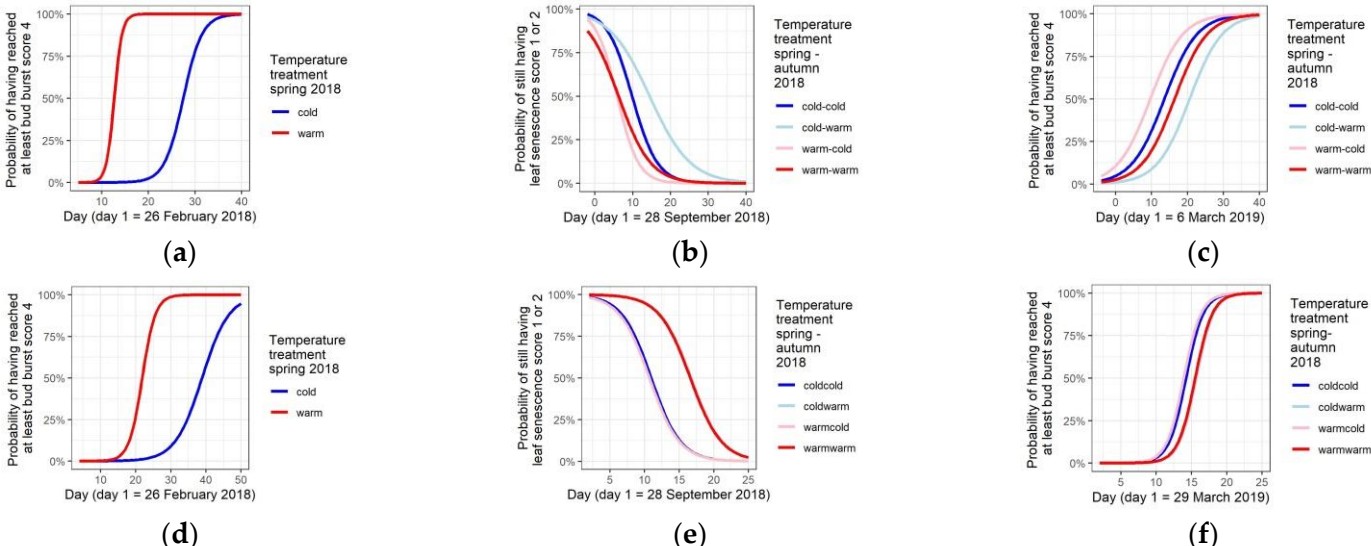

**Figure 4.** Modeled probability of having reached at least a given bud burst (**a**,**c**,**d**,**f**) or maximally a given leaf senescence (**b**,**e**), according to the temperature treatments in 2018. The modeled phenophases are shown for plants with an average height and belonging to the Belgian *Corylus avellana* for the *Corylus* taxa (**a**–**c**) and to the Belgian provenance for *Cornus sanguinea* (**d**–**f**). Bud burst is depicted for 2018 and 2019, while senescence is depicted for 2018.

Leaf senescence in 2018 was delayed in the *Cornus sanguinea* plants that experienced the warmer summer–autumn condition in 2018 ($p$ = 0.002 for the "cold–warm" condition, $p$ = 0.001 for the "warm–warm" condition, Table 4, Figure 4e). The duration of the leaf senescence in 2018 was not influenced by the temperature treatments in the *Cornus sanguinea* plants (no significant interaction terms between day and temperature treatments in Table 4). In the following spring of 2019, the seedlings from the warmer summer–autumn condition in 2018 displayed a delayed bud burst ($p$ = 0.001 for "cold–warm", $p$ < 0.001 for "warm–warm", Table 4, Figure 4f). No further after-effects were detected in the timing of the leaf senescence in 2019 or in the timing of the bud burst and the leaf senescence in 2020 (no significant $p$-values for the temperature treatments, Table 4).

### 3.3. Growth Differentiation among the Taxa and Provenances in the Common Garden

After the first growing season in 2017 (and before the temperature treatments in 2018), the seedlings from the Belgian intermediate forms, the Belgian *Corylus maxima*, and the Spanish *Corylus avellana* did not differ from the Belgian *Corylus avellana* for both plant height and stem diameter (no significant $p$-values for the taxa/region, Table S1). In 2018, the year of the temperature treatments, and the following 2 years, the height increments for the Belgian intermediate forms, the Belgian *Corylus maxima*, and the Spanish *Corylus avellana* also did not differ from the Belgian *Corylus avellana* (no significant $p$-values for the taxa/region, Table 3). This was likewise the case for the diameter increments in 2019 and 2020 (no significant $p$-values for the taxa/region, Table 3). Only a small differentiation could be detected for the diameter increment in 2018, with the Belgian intermediate forms and the Belgian *Corylus maxima* having a slightly larger diameter increment in this year ($p$ = 0.025 for the intermediate forms and $p$ = 0.026 for *Corylus maxima*, Table 3).

For *Cornus sanguinea*, the Spanish provenance displayed a lower height and a smaller diameter than the Belgian provenance in the first growing season of 2017 ($p$ < 0.001 for both height and diameter in Table S1). In the second growing season (2018), the height and diameter increments did not differ significantly between the Belgian and the Spanish provenance (no significant $p$-value for the Spanish provenance, Table 3). In 2019, there was a slightly smaller height increment for the Spanish provenance compared to the Belgian provenance, and, in 2020, this difference in height increment was stronger ($p$ = 0.047 in 2019

and *p* < 0.001 in 2020 for the Spanish provenance, Table 3), whereas the diameter increment did not differ between the two provenances in 2019 and 2020 (no significant *p*-value for the Spanish provenance, Table 3).

*3.4. Phenological Differentiation among the Taxa and Provenances in the Common Garden*

For the *Corylus* taxa, bud burst in 2018 occurred earlier in the Belgian *Corylus maxima* compared to the Belgian *Corylus avellana* (*p* = 0.008, Table 4, Figure 5a). The timing of leaf senescence in 2018 was slightly earlier in the Spanish *Corylus avellana* (*p* = 0.026, Table 4, Figure 4b). In 2019, the timing of the bud burst in the Belgian *Corylus maxima* and the Belgian intermediate forms was earlier than the Belgian *Corylus avellana* (*p* < 0.001 and *p* = 0.034 respectively, Table 4, Figure 5b). The picture for bud burst in 2020 was similar as in 2019 with the Belgian *Corylus maxima* and the Belgian intermediate forms being earlier than the Belgian *Corylus avellana* (*p* = 0.003 and *p* = 0.02 respectively, Table 4). For leaf senescence in 2020, the Belgian intermediate forms were slightly later than the Belgian *Corylus avellana* (*p* = 0.01, Table 4).

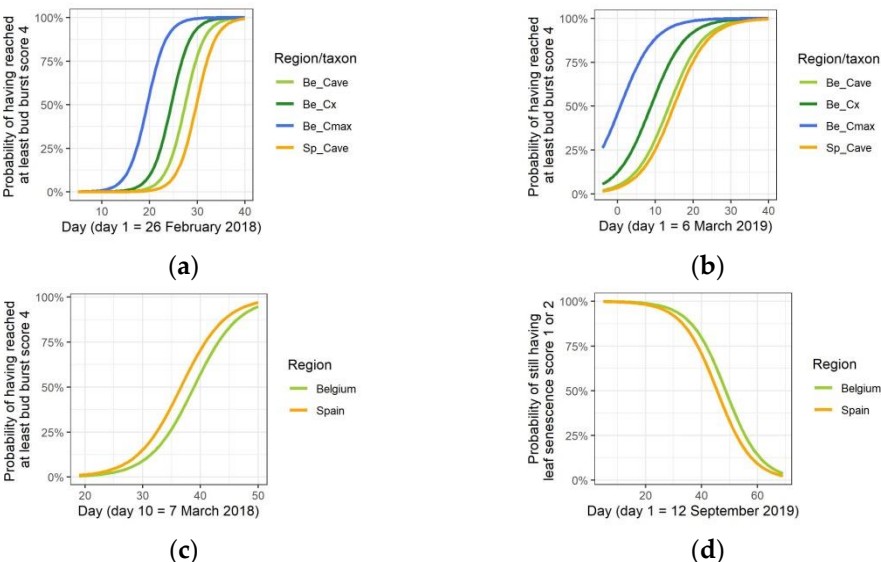

**Figure 5.** Modeled probability of having reached at least a given bud burst score (**a**–**c**) or maximally a given leaf senescence score (**d**), depending on the taxon/region for the *Corylus* taxa (**a**,**b**) and on the region for *Cornus sanguinea* (**c**,**d**). All model fits are for an average plant height and for the "cold–cold" temperature treatments in 2018. Abbreviations for the Corylus taxa are in Table 1.

In *Cornus sanguinea*, bud burst in the Spanish provenance started earlier than the Belgian provenance in 2018 (*p* = 0.017, Table 4, Figure 5c). In the following 2 years, bud burst was also earlier in the Spanish provenance (*p* < 0.001 for both 2019 and 2020, Table 4). There was no differentiation among the provenances for the timing of leaf senescence in 2018 and 2020. Only in 2019, leaf senescence was earlier in the Spanish provenance (*p* = 0.008, Table 4).

## 4. Discussion

*4.1. Responses to Spring and Summer–Autumn Warming*

Periodic warming in our experiment affected the two examined shrub species differently. In the *Corylus* taxa, a warmer spring temperature created a stress reaction, as growth was reduced in this year. Remarkably, the reduced diameter increment in the year of the warming treatments was restored with full compensation in the following year (the warm spring temperature plants displayed a larger diameter increment than the control plants). Height increment did not differ from the control plants in the year after the warming treatments, which is a restoration of growth without compensation. This growth response

for the diameter increment in *Corylus* can be related to the response of *Fagus sylvatica* L. in a spring warming experiment [36]. Here, both height increment and diameter increment displayed a similar reduction upon spring warming, expressing a stress reaction. The beech plants restored growth with a full compensation for height increment in the following year and for diameter increment 2 years later [36]. In oak, compensation growth has been described after drought stress [37,38].

The response of the *Cornus sanguinea* seedlings deviated from the *Corylus* taxa, with results pointing to a stronger sensitivity for summer–autumn warming. Height increment was smallest in the treatment year for the *Cornus sanguinea* seedlings that experienced the cold summer–autumn warming ("cold–cold" and "warm–cold" conditions). The "cold–warm" condition resulted in the largest height increment in this year. The larger height increment in the warmer summer–autumn condition may be, at least partly, explained by the later leaf senescence observed on these plants, implying a longer growing season. From in situ observations, it is known that higher autumnal temperatures can delay autumnal senescence [39]. Likewise, experimental warming can delay autumnal bud formation and leaf senescence as, e.g., has been reported for *Populus tremula* L. [40] or *Larix principis-rupprechtii* Mayr [21]. In addition, plant height and height increment correlated very poorly with timing of bud burst, but strongly with timing of bud set and leaf senescence in *Populus trichocarpa* L., as observed in transplant experiments [41], suggesting a mechanism in which height growth is relatively more increased by a delayed autumnal senescence than by an earlier bud burst. This corroborates other indications that sensitivity of the timing of autumnal leaf senescence to temperature may more strongly control the length of the growing season and plant productivity than the timing of bud burst in spring [42–44].

The effect of a delayed leaf senescence in 2018 on the diameter increment in *Cornus sanguinea* is less evident than on the height increment. Only the "cold–warm" condition resulted in an increased increment in this year, which was still evident in the year after the temperature treatments, with even a small increase 2 years later. This can be related to beech seedlings that were stressed by an elevated temperature, displaying a compensation growth in diameter up to 1 year later than the compensation growth in height [36], and may lead to the hypothesis that radial growth may hold the capacity to "remember" stressful events for a longer time in comparison to height growth. This can also be related to the finding in Scotch pine where radial growth is able to maintain its growth rate during an unfavourable cool year while height declines [45]. The fact that the cold spring does not lead to a different diameter increment than the warm spring, when combined with a cold summer–autumn temperature, whereas the cold spring does allow a larger diameter increment than the warm spring when combined with a warm summer–autumn, points to the spring temperature mediating the response on the summer–autumn temperature for diameter growth, thus demonstrating that the effects of spring and summer–autumn warming on radial growth are not additive but interact. The warmer spring temperature likely created a stress reaction in the plants which was not visible in the radial increment of the cold summer–autumn condition but reducing the stimulating effect of the warm summer–autumn in the "warm–warm" condition compared to the "cold–warm" condition.

Contrary to *Cornus sanguinea*, a delay of autumnal leaf senescence upon summer-autumn warming was not strongly expressed in the *Corylus* taxa, whereas the duration of this phenophase was clearly longer (which was not the case in *Cornus sanguinea*). Thus, the same elevated summer–autumn temperature can result in a delay of the onset of leaf senescence or in a longer duration of the phenophase, in a species-dependent way. Furthermore, a combination of these two effects upon summer–autumn warming has been observed in *P. spinosa* [31].

Together, our results for the increment growth upon periodic warming suggest that the *Corylus* taxa are more sensitive to spring warming, influencing growth negatively (height and diameter), whereas *Cornus sanguinea* seems more sensitive to summer–autumn warming, influencing height growth positively. After-effects in the following year for growth are absent (*Corylus*) or limited (*Cornus*).

Variability of the photosynthesis–temperature responses among plant species may result from acclimation to temperature, which involves short- to long-term changes at the level of an organism, and from adaptation to temperature, which involves evolutionary changes in deviating growth environments [46]. The difference in response between the *Corylus* taxa and *Cornus sanguinea* that we observed could be, at least partly, explained by their slightly diverging ecological niches, with the *Corylus* taxa being more shade-tolerant compared to *Cornus sanguinea* [47]. This may imply that the optimum temperature for photosynthesis in the *Corylus* taxa may be adapted to a buffering, milder forest climate compared to *Cornus sanguinea*, resulting in the stress at higher temperatures in spring, as well as less responsiveness to warmer summer–autumn temperatures. This hypothesis could also account for the similar stress reaction upon high spring temperature in the more shade-tolerant beech [36].

### 4.2. Common Garden

The common garden setting allowed looking for genetic differences in growth and phenology among the different taxa in *Corylus* and among the Belgian and Spanish provenances for both *Corylus avellana* and *Cornus sanguinea*. It should be noted, however, that results from *Corylus maxima* were derived from a limited number of plants (Table 1).

In the *Corylus* taxa originating from Belgium, bud burst was earlier in *Corylus maxima* in the three observation years. For the intermediate forms, bud burst was slightly earlier in the second and third years. Therefore, it can be hypothesised that the intermediate forms may have inherited an earlier timing of bud burst from non-native genotypes of large fruited *Corylus avellana* and/or *Corylus maxima*. Both *Corylus maxima* and the intermediate forms did not differ from the autochthonous *Corylus avellana* for the timing of leaf senescence, with the exception of a small effect for the intermediate forms in the second year after the treatments (slightly earlier). In terms of growth, *Corylus maxima* and the intermediate forms displayed a slightly larger diameter increment only in the year of the treatments. In the following 2 years, there was no difference. For height increment, there was no difference in the 3 years of observation. Together, differentiation among the taxa can be found mainly in the timing of bud burst, corroborating the finding from phylogenetic research that hybridisation between *Corylus avellana* and *Corylus maxima* is quite probable [21].

The Spanish *Corylus avellana* did not differ from the Flemish *Corylus avellana* for the timing of bud burst and the timing of leaf senescence, with the exception of a slightly earlier leaf senescence for the Spanish provenance in the year of the treatments. Moreover, the height and diameter increments did not differ between the two provenances in the 3 years of observation. Although the climate at the Spanish origin is harsher in terms of mean minimum and maximum temperatures, this did not result in differentiation between the Spanish and the Belgian provenances for growth or phenology in the common garden setting. These observations allow hypothesizing that, for *Corylus avellana*, the strength of phenotypic plasticity as an adaptive strategy may be higher than genetic population differentiation for the studied phenological and growth traits.

From the 3 years of phenological observations in the common garden, it can be deduced that the growing season in the Spanish provenance is advanced in comparison to the Flemish provenance with an earlier bud burst and an earlier leaf senescence. The growth results indicate that the climatic harsher growth conditions at the home site of the Spanish provenance is expressed mainly in height growth retardation in the common garden setting, possibly due to the earlier leaf senescence having a greater negative impact on growth than the putative positive impact of an earlier bud burst in spring.

Although the *Corylus avellana* and *Cornus sanguinea* plants originated from similar origins, both in Belgium and in Spain, their responses in the common garden setting diverged, with *Corylus avellana* displaying more stable responses and *Cornus sanguinea* displaying more variation between the Belgian and Spanish provenances, suggesting diverse differentiation processes.

## 5. Conclusions

Our results show that the spring warming, the summer–autumn warming, and the combination of both affected height and diameter growth, with the *Corylus* taxa being more sensitive to spring warming, influencing growth negatively (height and diameter), and *Cornus sanguinea* being more sensitive to summer–autumn warming, influencing height growth positively. A stressful temperature elevation in spring caused growth retardation, followed by growth restoration with full compensation for diameter increment in *Corylus avellana*. For *Cornus sanguinea*, the higher sensitivity to autumnal elevated temperature corroborated the finding that autumnal warming with a concurrent delay of leaf senescence may have a strong effect on growth, even stronger than an earlier bud burst in spring. Results for this species also lead to the hypothesis that radial growth in seedlings and saplings may hold the capacity to "remember" stressful events for a longer time in comparison to height growth. In addition, analysis of the diameter increments in *Cornus sanguinea* also demonstrated that the effects of spring and summer–autumn warming are not additive but interact. The observed differences in responses upon the periodic warming between the *Corylus* taxa and *Cornus sanguinea* could result from the fact that the *Corylus* taxa are more shade-tolerant in comparison to *Cornus sanguinea*.

Our results also showed that the timing and the duration of the phenophases bud burst and leaf senescence were affected by the applied spring and/or summer–autumn warming. Results from the *Corylus* taxa indicate a putative impact of warming on the economically valuable hazelnut production in Europe, as altered timing of vegetative phenophases may influence not only growth but also the timing of reproductive phenophases. As warming advanced bud burst in our experiment, it may have also resulted in earlier flowering and fruiting. The putative impact of drought, which often co-occurs with higher temperatures, should be part of future research, together with the putative higher impact of late spring frosts which may happen more frequently as spring phenophases advance. Lastly, we found that differences in growth and phenology were expressed between the different *Corylus* taxa and between the local and nonlocal provenances. Both the *Corylus avellana* and the *Cornus sanguinea* provenances, originating from similar home-sites in Belgium and in Spain, showed divergent responses in the common garden, with *Corylus avellana* displaying more stable responses and *Cornus sanguinea* displaying more variation between the Belgian and Spanish provenance, suggesting diverse differentiation processes. Altogether, the results depicted in this study suggest that genetic differentiation and phenotypic plasticity, for the analyzed traits, are two powerful strategies in shrubs, similar to trees, to cope with environmental changes, and that they are species-specific. The quick recovery from stressful growth conditions, as observed in our experiment, showed the capacity of the *Corylus* taxa and *Cornus sanguinea* to face some consequences of climate change, not without conspicuous effects.

It should be noted that this study had its limitations. Firstly, results from seedlings and saplings in greenhouse experiments should only be extrapolated to field environments with caution. Secondly, the experiment was not repeated in space or time.

**Supplementary Materials:** The following supporting information can be downloaded at https://www.mdpi.com/article/10.3390/f13050654/s1: Table S1. Estimates and *p*-values for the modeled height and diameter of the *Corylus* taxa and of *Cornus sanguinea* at the end of 2017. For the *Corylus* taxa, the standard taxon/region is the Belgian *Corylus avellana* to which the Belgian intermediate forms (Be_Cx), the Belgian *Corylus maxima* (Be_Cmax), and the Spanish *Corylus avellana* (Sp_Cave) are compared. For *Cornus sanguinea*, the Belgian provenance is the standard provenance to which the Spanish provenance (Sp) is compared.

**Author Contributions:** Conceptualization of experiment, K.V.M., J.M.M. and E.N.P.; methodology, K.V.M., J.M.M., S.M., E.N.P. and A.T.; measurements and observations, J.M.M. and S.M.; validation of data, K.V.M. and J.M.M.; statistical analysis, K.V.M. and J.M.M.; preparation of draft manuscript, K.V.M. and A.T.; editing and reviewing of the manuscript, K.V.M., A.T., E.N.P. and J.M.M. All authors have read and agreed to the published version of the manuscript.

**Funding:** This research received no external funding.

**Data Availability Statement:** Data are available at 10.5281/zenodo.6141015.

**Acknowledgments:** We thank Joris De Wolf for his help with seed collection. We also thank Lisa Carnal, Amy Lauwers, Cédric Van Dun, Yorrick Aguas Guerreiro, Segolene Bauduin, Stijn De Leenheer, Denis Cattoir, and Matthieu Gallin for contributing to the data acquisition. Marc Schouppe and Nico De Regge were great in taking care of the plants.

**Conflicts of Interest:** The authors declare no conflict of interest.

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
