# Peer review of "Stimulation, Reduction and Compensation Growth, and Variable Phenological Responses to Spring and/or Summer–Autumn Warming in Corylus Taxa and Cornus sanguinea L."

_forests, doi:10.3390/f13050654_

Round 1
Reviewer 1 Report
Dear authors, I find this paper interesting. There are some revisions needed, which are detailed below:
Abstract:
I suggest adding phenology results in the Abstract.
Introduction:
In the first paragraph, I suggest to add more recent references, as well as the IPCC newest report information about climate change and its effects on ecosystems.
In the second paragraph, I suggest to mention that specially for species with economical uses (as Corylus taxa), phenological shifts due to rising temperatures may alter vegetative phenophases and, consequently, fruit production and impact on crop production.
Line 71: “en”? I guess authors mean “and”
Lines 71-86: please, add common phenological timings for leaf fall, flush, flowering and fruiting of each of 3 species. Also, please, add description of fruit morphology of Cornus sanguinea. Additionally, authors attest that Corylus avellana and Cornus sanguinea occur naturally, but are they dense and common species? Please add information about density.
Line 89: what is a regular shrub species? Maybe the population structure description would help here. Also, again, Corylus avellana is an economically important species and that information should be more explored, as there might be further implications for crop production.
Lines 92-95: I suggest to transfer that to Methods.
What were the expectations or hypothesis?
Methods:
Table 1 is cited before Figure 1, but presented later.
Line 108: Authors argue that “In selecting Corylus avellana populations, nuts were carefully chosen to ensure that small nuts with not too long involucres were collected”. I understand that. However, generally, small nuts mean lower seed vigour which affects seedling fitness and growth. Please, I suggest adding information regarding if seed size of all three species could have affected results.
In Table 1, as much as I think results of Corylus maxima are important, they come from a limited number of seedlings and from an even more limited number of mother plants. I suggest authors make clear statements about that or remove this species of the study.
Line 110-113: is there any reference to help to describe intermediate forms?
In Figure 1, I suggest adding climate information for the intermediate forms and Corylus maxima.
How seeds were cleaned?
I suggest: Figure 2. Mean daily temperatures in the greenhouse chambers in spring (a) and in summer-autumn (b) in 2018.
Line 148: is 5.6 C a significant temperature difference? Authors could use IPCC reports to justify that.
Line 161: and what about 1.9 C? is that a significant temperature difference? Please, justify that.
If I understood correctly, temperature treatments happened only in 2018 and measurements were conducted till 2020. Is that correct? Why would you expect such a long effect? I suggest adding that information.
Line 181: how regularly?
Is there a reason for the changes in the beginning and end of observation dates among years in Table 2? I am asking this, because studies of heating effects on reproductive phenology are showing species to flower and fruit earlier, when submitted to higher temperatures, and that might be an interesting information that your study brings.
Results:
In Table 3, where are the cold-cold results?
In Table 4, what is Tw?
In Figure 5, please provide full captions.
I suggest Discussion to be less descriptive. Some parts seemed like a repetition of the Results. I suggest adding more comparative studies, adding the perspective of how the results could be interesting to a broad audience and the importance of the results to the context of climate change.
Conclusions should bring back the hypothesis and confirm or not them.
I hope this review helps.
Author Response
Dear authors, I find this paper interesting. There are some revisions needed, which are detailed below:
Abstract:
I suggest adding phenology results in the Abstract.
We added: “The warm spring condition advanced bud burst in all studied taxa, whereas the warm summer-autumn advanced leaf senescence in the Corylus taxa and delayed it in Cornus sanguinea. Little to no after effects of the temperature treatments were detected.”
Introduction:
In the first paragraph, I suggest to add more recent references, as well as the IPCC newest report information about climate change and its effects on ecosystems.
We have added now several recent references and the IPCC 2022 report to the first paragraph:
IPCC. Climate Change 2022: Impacts, Adaptation, and Vulnerability. Contribution of Working Group II to the Sixth Assessment Report of the Intergovernmental Panel on Climate Change; Pörtner, H.-O., Roberts, D.C., Tignor, M., Poloczanska, E.S., Mintenbeck, K., Alegría, A., Craig, M., Langsdorf, S., Löschke, S., Möller, V., et al., Eds.; Cambridge University Press, 2022.
Chung, H.; Muraoka, H.; Nakamura, M.; Han, S.; Muller, O.; Son, Y. Experimental warming studies on tree species and forest ecosystems: a literature review. Journal of plant research 2013, 126, 447-460, doi:10.1007/s10265-013-0565-3.
Malyshev, A.V. Warming Events Advance or Delay Spring Phenology by Affecting Bud Dormancy Depth in Trees. Frontiers in Plant Science 2020, 11, doi:10.3389/fpls.2020.00856.
Zohner, C.M.; Renner, S.S. Ongoing seasonally uneven climate warming leads to earlier autumn growth cessation in deciduous trees. Oecologia 2019, 189, 549-561, doi:10.1007/s00442-019-04339-7.
Beil, I.; Kreyling, J.; Meyer, C.; Lemcke, N.; Malyshev, A.V. Late to bed, late to rise-Warmer autumn temperatures delay spring phenology by delaying dormancy. Glob Chang Biol 2021, 27, 5806-5817, doi:10.1111/gcb.15858.
In the second paragraph, I suggest to mention that specially for species with economical uses (as Corylus taxa), phenological shifts due to rising temperatures may alter vegetative phenophases and, consequently, fruit production and impact on crop production.
We added: “In addition, phenological shifts due to climate change may not only alter vegetative phenophases but also reproductive phenophases, impacting crop production for woody species producing fruits with an economic value as e.g. Corylus taxa [11].”
Line 71: “en”? I guess authors mean “and”
Indeed. We have changed this.
Lines 71-86: please, add common phenological timings for leaf fall, flush, flowering and fruiting of each of 3 species.
Local natural autochthonous populations of shrub and tree species are adapted to local growth conditions. It is known that timing of e.g. leaf flushing is dependent on this local adaptation (temperature, precipitation, photoperiod,…), in interplay with the prevalent meteorological conditions and microclimate. So, it is not so easy to extract general descriptions (absolute dates) for the timing of different phenophases at the level of a species, as may be the case for selected and breeded clonal crop varieties. It is e.g. also known that within a natural population a large variability is present in timing of a given phenophase. It is not the purpose of this paper to look at absolute dates for the timing of the studied phenophases but rather to look at their relative change (comparison between cold and warm conditions) upon warming. But, maybe we misunderstood the suggestion of the reviewer?
Also, please, add description of fruit morphology of Cornus sanguinea.
We added: “The non-toxic but unpleasant tasting fruits of C. sanguinea are berry-like drupes, 5 – 8 mm wide, without any economic value [18]. “
Additionally, authors attest that Corylus avellana and Cornus sanguinea occur naturally, but are they dense and common species? Please add information about density. Line 89: what is a regular shrub species? Maybe the population structure description would help here.
For Corylus avellana we added: “In natural stands, it is widely distributed in Europe, ranging from Scandinavia to the south of the continent [19]. Also in Belgium, the species is very common [20].”
And for Cornus sanguinea: “Also in Belgium, the species is very common [20].”
Also, again, Corylus avellana is an economically important species and that information should be more explored, as there might be further implications for crop production.
We added: “The production of edible hazelnuts is an important economic activity, as e.g. in Turkey [27].”
Lines 92-95: I suggest to transfer that to Methods.
Unfortunately, we had to leave the sentences in, as this information on the experimental set-up is necessary to understand the hypotheses that follow.
What were the expectations or hypothesis?
We have formulated the hypotheses: “We hypothesized that (i) a spring warming, a summer-autumn warming or a combination of both, affect height and diameter growth, that (ii) the timing and duration of the phenophases bud burst and leaf senescence are affected by the applied spring and/or summer-autumn warming, and that (iii) differences in growth and phenology are expressed between the different Corylus taxa and between the local and non-local provenances.”
Methods:
Table 1 is cited before Figure 1, but presented later.
We moved Table 1 up now.
Line 108: Authors argue that “In selecting Corylus avellana populations, nuts were carefully chosen to ensure that small nuts with not too long involucres were collected”. I understand that. However, generally, small nuts mean lower seed vigour which affects seedling fitness and growth. Please, I suggest adding information regarding if seed size of all three species could have affected results.
If the larger nuts of Corylus maxima and the intermediate forms would lead to more vigorous seedlings in comparison to Corylus avellana, then the height of the seedlings after the first growing season would differ between these taxa, which is not the case. In the first paragraph of 3.3 (Growth differentiation among the taxa and provenances in the common garden) we write : “After the first growing season in 2017 (and before the temperature treatments in 2018), the seedlings from the Belgian intermediate forms, the Belgian Corylus maxima and the Spanish Corylus avellana did not differ from the Belgian Corylus avellana for both plant height and stem diameter (no significant p-values for the taxa/region, Table S1).” Therefore, we thought it wise not to add more information on the possible influence of nut size on the presented results, mainly to avoid confusion.
In Table 1, as much as I think results of Corylus maxima are important, they come from a limited number of seedlings and from an even more limited number of mother plants. I suggest authors make clear statements about that or remove this species of the study.
We added in paragraph 4.2. (Common garden): “It should be noted though that results from Corylus maxima are derived from a limited number of plants (Table 1). “
Line 110-113: is there any reference to help to describe intermediate forms?
Two references are present in the Introduction: “…Corylus avellana has for long been selected and cultivated for larger nuts, and intermediate forms that probably originated from spontaneous crosses between cultivated and natural Corylus avellana have been described [25,26].”
In Figure 1, I suggest adding climate information for the intermediate forms and Corylus maxima.
We added in the text: “Corylus maxima and the intermediate forms were also collected in Belgium, implying that the Belgian climate data in Figure 1 count for these two taxa as well. “ We added in the caption of Figure 1: “Climate and day length at the origin of the provenances of Corylus avellana, Corylus maxima and the intermediate forms, and Cornus sanguinea.”
How seeds were cleaned?
We added: “Berries were soaked overnight in water and seeds were manually pushed out from the berries.”
I suggest: Figure 2. Mean daily temperatures in the greenhouse chambers in spring (a) and in summer-autumn (b) in 2018.
We have adjusted the caption as suggested.
Line 148: is 5.6 C a significant temperature difference? Authors could use IPCC reports to justify that.
Line 161: and what about 1.9 C? is that a significant temperature difference? Please, justify that.
As we worked in greenhouse conditions, not in growth chambers, the temperature differences were not clear-cut, but depended on the outdoor climate (temperature, hours of sunshine,…). Our objective was to look at relative differences in responses between a warmer and a colder condition, not to look at responses upon a given, strict temperature regime. In the last paragraph of the introduction, before the hypotheses, we added: “The experiment was conducted in greenhouse conditions. We aimed to look at relative differences in plant responses.”
If I understood correctly, temperature treatments happened only in 2018 and measurements were conducted till 2020. Is that correct? Why would you expect such a long effect? I suggest adding that information.
Sometimes, temperature treatments may display effects up to two years after the treatment, as was detected in Fagus sylvatica. This is mentioned in the discussion (4.1. Responses to spring and summer-autumn warming): “…can be related to the response of Fagus sylvatica L. in a spring warming experiment [32]. Here, both height increment and diameter increment displayed a similar reduction upon spring warming, expressing a stress reaction. The beech plants restored growth with a full compensation for height increment in the following year and for diameter increment two years later”
Line 181: how regularly?
We are not sure if we understand the question correctly. In the sentence: “The rate of bud burst was observed regularly in the spring of 2018, 2019 and 2020 (Table 2),…” we mention Table 2. In Table 2 all observation dates are clearly indicated.
Is there a reason for the changes in the beginning and end of observation dates among years in Table 2? I am asking this, because studies of heating effects on reproductive phenology are showing species to flower and fruit earlier, when submitted to higher temperatures, and that might be an interesting information that your study brings.
We indeed looked at the change in phenology upon warming. In the sections on phenology we show how higher temperatures influence the phenophases. As this concerns a greenhouse experiment, the absolute number of days of the phenological shifts is not so relevant. It was not our purpose to link exact temperature regimes with the size of the phenological shifts. As the temperature regimes inside the greenhouse and outdoors differ, and because of inter-annual climate variability, the dates of the starting and ending of the observations of the phenophases differ between the years of observation. It was not our purpose to explain these absolute inter-annual differences.
Results:
In Table 3, where are the cold-cold results?
It is mentioned in the caption: “The “cold-cold” condition in the temperature treatments during 2018 is the standard to which the other three conditions “cold-warm”, “warm-cold” and “warm-warm” are compared (Tcw, Twc and Tww respectively).”
In Table 4, what is Tw?
We added in the caption: “For the bud burst in 2018 there were only two conditions yet, with the “cold” condition as standard to which the “warm” condition is compared to (Tw).”
In Figure 5, please provide full captions.
We are not sure what is asked for here. We added in the caption: “Abbreviations for the Corylus taxa are in Table 1.”
I suggest Discussion to be less descriptive. Some parts seemed like a repetition of the Results. I suggest adding more comparative studies, adding the perspective of how the results could be interesting to a broad audience and the importance of the results to the context of climate change.
We have deleted many parts in the discussion and the conclusion, to make the text less repetitive of the results and less descriptive. We added a paragraph in the conclusions, to make the text more interesting to a broader audience: “Our results also showed that the timing and the duration of the phenophases bud burst and leaf senescence were affected by the applied spring and/or summer-autumn warming. Results from the Corylus taxa indicate a putative impact of warming on the economically valuable hazelnut production in Europe as altered timing of vegetative phenophases may influence not only growth but also the timing of reproductive phenophases. As warming advanced bud burst in our experiment, it may also results in earlier flowering and fruiting. The putative impact of drought, which often co-occurs with higher temperatures, should be part of future research, together with the putative higher impact of late spring frosts which may happen more frequently as spring phenophases advance.”
Conclusions should bring back the hypothesis and confirm or not them.
We re-arranged the conclusions so that the different hypothesis are dealt with. The three formulated hypotheses are tackled in the three paragraphs of the conclusions.
I hope this review helps.
It did, indeed!
Reviewer 2 Report
Species-specific response to climate change represents a critical area of research, as we seek to fill in knowledge gaps and develop models of plant growth, development, and reproduction in the face of a changing climate. This article is well written and the content is well within the scope of this journal, and should be of broad interest.
I did not see (but may have missed) a section on limitations of this study. The authors should note the limitations of extrapolating data derived from growth chamber and greenhouse studies to field-grown trees. The general approach followed is fine, but please add a sentence or two on these limitations.
In line 506ff, the authors hypothesized about genetic inheritance of earlier bud break. This is certainly possible, but I will note that epigenetic control is also likely. I am not as familiar with these species, but in at least a few other species, epigenetic control of bud break based on the environmental conditions of the parent plants during pollination and seed development has been documented. This does not need to be added to this manuscript, but I wanted to point this out in case it is of interest to the authors.
I do have serious concerns about the statistical design. Based on my reading, it appears that temperature treatments were not replicated. A minimum of 3 growth chambers or greenhouses should be used for each treatment, allowing a statistical comparison between treatments. Without that, the authors can really only compare within each treatment group, not across treatments. This study should have been replicated either in the manner described above or over time, repeating the experiment with new plants in subsequent years. However, I may have misunderstood the design, in which case I welcome the authors’ response. Also, I do understand that papers are routinely published these days without true replication, and I will leave it up to the editors to decide whether it may be published as is. If the editors find it acceptable to publish without true replication, then the authors should minimally be asked to write a few sentences explaining how lack of replication limits interpretability of the results.
Similarly, in the Results section, the authors compare (lines 259f and 277f) growth differences between Corylus taxa and C. sanguinea. Because these were not compared statistically, these sentences should be removed. It is fine to include them under discussion, but these are not results of the study. In the Results section, only results of statistical tests should be reported, not the authors’ interpretation of those results.
Author Response
Species-specific response to climate change represents a critical area of research, as we seek to fill in knowledge gaps and develop models of plant growth, development, and reproduction in the face of a changing climate. This article is well written and the content is well within the scope of this journal, and should be of broad interest.
I did not see (but may have missed) a section on limitations of this study. The authors should note the limitations of extrapolating data derived from growth chamber and greenhouse studies to field-grown trees. The general approach followed is fine, but please add a sentence or two on these limitations.
In the conclusions we added: “It should be noted that this study has its limitations. First, results from from seedlings and saplings in greenhouse experiments should only be extrapolated to field environments with caution. Secondly, the experiment was not repeated in space nor in time.”
In line 506ff, the authors hypothesized about genetic inheritance of earlier bud break. This is certainly possible, but I will note that epigenetic control is also likely. I am not as familiar with these species, but in at least a few other species, epigenetic control of bud break based on the environmental conditions of the parent plants during pollination and seed development has been documented. This does not need to be added to this manuscript, but I wanted to point this out in case it is of interest to the authors.
We are aware of epigenetic control of phenological events, the first author has been involved in these type of studies. The experimental set-up of the here described study does not allow drawing conclusions about epigenetic influences.
I do have serious concerns about the statistical design. Based on my reading, it appears that temperature treatments were not replicated. A minimum of 3 growth chambers or greenhouses should be used for each treatment, allowing a statistical comparison between treatments. Without that, the authors can really only compare within each treatment group, not across treatments. This study should have been replicated either in the manner described above or over time, repeating the experiment with new plants in subsequent years. However, I may have misunderstood the design, in which case I welcome the authors’ response. Also, I do understand that papers are routinely published these days without true replication, and I will leave it up to the editors to decide whether it may be published as is. If the editors find it acceptable to publish without true replication, then the authors should minimally be asked to write a few sentences explaining how lack of replication limits interpretability of the results.
We have added this limitation of the study in the conclusions section as mentioned above: “It should be noted that this study has its limitations. First, results from from seedlings and saplings in greenhouse experiments should only be extrapolated to field environments with caution. Secondly, the experiment was not repeated in space nor in time.”
Similarly, in the Results section, the authors compare (lines 259f and 277f) growth differences between Corylus taxa and C. sanguinea. Because these were not compared statistically, these sentences should be removed. It is fine to include them under discussion, but these are not results of the study. In the Results section, only results of statistical tests should be reported, not the authors’ interpretation of those results.
We deleted in the results section:
“At the end of the two temperature treatments in 2018, there was a marked difference in height and diameter increments between the Corylus taxa and Cornus sanguinea. …
The seedlings of Cornus sanguinea exhibited a deviating response to the temperature treatments in comparison to the Corylus taxa. …
In the second temperature treatment in the summer-autumn of 2018, with four conditions “cold-cold”, “cold-warm”, ”warm-cold” and “warm-warm”, the Corylus taxa and Cornus sanguinea responded differently. …
In contrast to the Corylus taxa, …“